**Data Availability Statement:** All relevant data are within the paper and its Supporting Information files.

# Retrospective survey of youth sports participation: Development and assessment of reliability using school records

Steven Jin [ID]¹, Amanda R. Rabinowitz²*, Jordan Weiss³, Sameer Deshpande⁴, Nitika Gupta⁵, Reuben A. Buford May⁶, Dylan S. Small⁷

1 College of Arts and Sciences, University of Pennsylvania, Philadelphia, Pennsylvania, United States of America, 2 Moss Rehabilitation Research Institute, Elkins Park, Pennsylvania, United States of America, 3 Population Studies Center and the Leonard Davis Institute of Health Economics, University of Pennsylvania, Philadelphia, Pennsylvania, United States of America, 4 Computer Sciences and Artificial Intelligence Library, Massachusetts Institute of Technology, Cambridge, Massachusetts, United States of America, 5 University of Pennsylvania School of Dental Medicine, Philadelphia, Pennsylvania, United States of America, 6 Department of Sociology, University of Illinois at Urbana-Champaign, Urbana, Illinois, United States of America, 7 Department of Statistics, The Wharton School, University of Pennsylvania, Philadelphia, Pennsylvania, United States of America

* RabinowA@einstein.edu

## Abstract

Many youths participate in sports, and it is of interest to understand the impact of youth sports participation on later-life outcomes. However, prospective studies take a long time to complete and retrospective studies may be more practical and time-efficient to address some questions. We pilot a retrospective survey of youth sports participation and examine agreement between respondent's self-reported participation with high school records in a sample of 84 adults who graduated from high school between 1948 and 2018. The percent agreement between our survey and the school resources for individual sports ranged between 91.5% and 100%. These findings provide preliminary evidence for the reliability of retrospective self-report of youth sports participation. This survey may serve as an efficient approach for evaluating relationships between involvement in youth sports and health outcomes later in adulthood.

## Introduction

Playing high school sports has historically been and continues to be a prominent part of adolescence and young adulthood. In the United States, nearly eight million students played a high school sport during the 2018–2019 school year [1]. However, participation in high school sports has recently declined (from 7,980,886 in 2017–2018 to 7,937,491 in 2018–2019), led by declines in football participation, marking the first decline in participation for more than 30 years [1]. Reasons for this decline may be multifaceted and partially attributed to growing concern over the short- and long-term implications on mental and cognitive health of playing collision sports [2]. State-level appropriations for public school athletic programs were also reduced in response to the Great Recession [3], which, some have argued, is justified in

**Funding:** SJ was supported by the Penn Undergraduate Research Mentoring program (https://www.curf.upenn.edu/content/penn-undergraduate-research-mentoring-program-purm) and the Wharton Dean's Research Fund. NG was supported by the Leonard Davis Institute of Health Economics SUMR Scholars program (https://ldi.upenn.edu/education/penn-ldi-training-programs/sumr/). JW received funding from the Population Research Training Grant (NIH T32 HD007242) awarded to the Population Studies Center at the University of Pennsylvania by the NIH's Eunice Kennedy Shriver National Institute of Child Health and Human Development (https://www.nichd.nih.gov/). The funders had no role in study design, data collection and analysis, decision to publish, or preparation of the manuscript.

**Competing interests:** The authors have declared that no competing interests exist.

response to potential deleterious effects on health and detracting from academic achievement [4]. To date, however, numerous studies that examined associations between participation in school sports and health or academic achievement found no association [5,6] while others reported that playing high school sports may have positive effects on educational attainment and future labor outcomes [7,8]. Additionally, many papers have reported positive associations between sports participation and physical fitness, brain function [9,10], and social and emotional health [11]. Meanwhile, a growing but mixed body of work links sports-related head trauma to mental and cognitive health deficits, prompting concern about the potential risks of adolescent participation in contact sports [12].

An ideal study of the long-term health implications of playing high-school sports would involve the collection of detailed, prospective information on sports participation with longitudinal follow-up that includes measures of cognitive and psychosocial functioning.

However, longitudinal studies of this magnitude are costly and the scientific community would have to wait decades for study participants to age to a point when the long-term risks of contact sports may manifest [13]. An alternative approach is to capitalize on existing longitudinal studies that collect both retrospective and prospective data, such as the National Longitudinal Study of Adolescent to Adult Health (Add Health). These large-scale studies are nationally representative and contain data spanning economic, health, and psychosocial outcomes.

The prospective identification of brain injury by observation or clinical diagnosis, is the gold standard method for injury ascertainment. This method is often too costly or unfeasible in many research designs. Fortunately, retrospective self-report measures have been developed and validated to collect incidence of prior head injury, such as the OSU-TBI ID [14] and the BISQ [15]. Data support that these instruments are valid and reliable indicators of prior brain trauma, including mild traumatic brain injury (mTBI) with associated loss of consciousness. Such measures have already been integrated into survey studies of long-term health. However, due to the absence of a known reliable survey to assess early-life sports participation, studies have been limited in their collection of information related to sports participation in early-life. This hampers the ability to contextualize possible deleterious effects of head injury against the backdrop of physical, emotional, and social benefits of youth sports participation.

To fill this gap, this study introduces a retrospective sports participation questionnaire administered electronically to adults who graduated from high school between 1948 and 2018. We evaluate the reliability of the questionnaire with respect to high school sports participation and find it to be reliable. Retrospective assessment of youth sports participation has the advantage of allowing investigators to capitalize on existing longitudinal datasets that have already tracked cognitive and health outcomes into later-life when some of the deleterious effects of brain injury are thought to manifest. Our survey represents a feasible and cost-effective approach for obtaining retrospective sports participation data.

## Materials and methods

### Survey development

Survey items were developed by the authors. Items were designed to capture information about the duration and intensity of exposure (i.e., sports participation). Intensity was operationalized as both weekly hours of participation and the competitive level of the team. A draft version of the survey was implemented using the Qualtrics platform for web-based survey administration. An electronic link to the survey was shared with two athletic trainers for comment after which the survey was piloted among a group of undergraduate research assistants (RAs) who were not involved in survey development. Feedback from that process was

incorporated, and the survey was updated for content, clarity, and ease of use. The finalized version was sent back to RAs for quality assurance before deployment in the study sample.

## Survey

In our survey, subjects were asked to select the sports in which they participated from a list of radio buttons. The survey used a series of blocks to assess sports participation at various stages of one's education. This included a block for grades K-6, grades 7–8, grades 9–12, and college. For each block corresponding to a specific period in the life course, respondents were asked to select the sports in which they participated from a list of 15 radio buttons. The specific 15 sports (listed in Table 1) included in our survey were selected based on sports participation surveys conducted by the National Federation of State High School Associations as well as knowledge of other sports that were popular at the schools where the pilot study described in the next section was conducted. Based on each respondent's selections, binary variables were constructed for participation in each sport during high school. Respondents also indicated whether, at each specific period, they had any physical limitations that prevented them from participating in sports. For each selected sport at each time period, respondents were asked to report (using radio buttons) the specific grades in which they participated, whether their team (a) traveled for games or tournaments, (b) participated in playoffs, (c) won a championship, or (d) none of the above. Respondents also indicated how many hours per week (0 hours, 1–4 hours, 5–9 hours, 10–14 hours, 15–19 hours, 20+ hours) they spent practicing and playing games for each specific sport, in addition to the position(s) they played. Our survey is provided in the supplementary materials.

## Sample collection

A pilot study was conducted to assess the reliability of the survey by constructing a convenience sample that distributed the survey through alumni e-mail lists and social media groups from high schools where authors (D.S. and A.R) are alumni. The volunteer sample consisted of all alumni from these two high schools who responded to a request to fill out the survey.

**Table 1. Percent agreement and internal consistency for survey responses and school resources.**

| Sport | Percent Agreement | Self-reported Participation | |
|---|---|---|---|
| | | Total | % |
| Football | 98.8 | 10 | 12.2 |
| Basketball | 92.7 | 22 | 26.8 |
| Baseball | 91.5 | 10 | 12.2 |
| Soccer | 96.3 | 24 | 29.3 |
| Lacrosse | 91.5 | 15 | 18.3 |
| Wrestling | 98.8 | 9 | 11 |
| Cross Country | 93.9 | 10 | 12.2 |
| Track and Field | 92.7 | 2 | 2.4 |
| Swimming | 95.1 | 15 | 18.3 |
| Softball | 97.6 | 9 | 11 |
| Volleyball | 98.8 | 6 | 7.3 |
| Tennis | 96.3 | 9 | 11 |
| Golf | 97.6 | 3 | 3.7 |
| Cheerleading | 96.3 | 1 | 1.2 |
| Bowling | 100 | 2 | 2.4 |

These high schools offered an advantage in that they maintain yearbook and/or online database records of student sports participation. The availability of such data from these high schools provided a straightforward method for evaluating the accuracy of self-reported high school sports participation data. Yearbook data were reviewed by two independent RAs on site at the individual schools. RAs abstracted data on sports participation by examining physical yearbooks or reviewing information from digital archives. The inspection of yearbooks involved examining all possible sources of sports participation information, including team photographs, rosters, and descriptions of extracurriculars for individual students. When multiple archival sources were available, all information was considered for the purposes of coding sport participation.

Disagreements in coding were rare, but when they occurred, they were resolved by consensus with input from all authors. The Institutional Review Board of the University of Pennsylvania reviewed the procedures of this study and determined that the study met eligibility criteria for IRB review exemption. Because the IRB judged the risk to be minimal to participants and the risk/benefit ratio to be reasonable, the IRB approved our process of providing the information about how the survey would be used to participants and not requiring explicit consent.

## Statistical analysis

We examined the agreement between respondent's self-reported sports participation and the information obtained from yearbooks/online databases by examining the percentage of survey responses that were in agreement with these external sources. For each individual sport, there were four possible outcomes: 1) the survey respondent reported participation that was confirmed by the school record, 2) the survey respondent reported participation but it *was not* confirmed by the school record, 3) the survey respondent did not report sports participation but there was a record of sports participation in the school yearbook/database, or 4) the survey respondent and the school record did not report participation. We removed cases from analysis if they could not be validated due to irregularities with the school resources available to us, e.g. if a sport was not included in the school record for a given year.

## Results

Eighty-two respondents completed the survey. The average time it took to complete the survey was 15.8 minutes (SD = 23.7 minutes). Modal completion time for the full sample was 6 minutes. The mean age of the sample was 42.2 years (SD = 14.4y); 61% (n = 50) of the sample was male and 83% (n = 68) of respondents were white. Respondents reported playing an average of 2.02 (SD = 1.09) sports.

The percent agreement between our survey and the school resources for individual sports ranged between 91.5% and 100% (Table 1). Table 1 also displays the frequency of self-reported participation for each sport.

To illustrate the duration and intensity information gathered through the survey, we also report data for several measures of intensity, such as years of play, weekly hours spent participating in practice and games, and team achievements (i.e. playoff appearances and championships won) for the most frequently participated in sports—basketball and soccer (Table 2).

## Discussion

In this paper, we introduce a new survey for gathering retrospective data on youth-sports participation. Comparisons of participants' responses with high school yearbook data suggested that most respondents were able to reliably report their high school athletic experiences. Out of 15 sports evaluated, agreement between survey data and school records was over 90% for all

**Table 2. Sports participation duration and intensity.**

|  | Basketball (N = 17) | Soccer (N = 20) |
|---|---|---|
| Years spent playing sport |  |  |
| Median | 3 | 6 |
| Minimum | 1 | 1 |
| 1st Quartile | 1 | 2 |
| 3rd Quartile | 5 | 11 |
| Max | 12 | 13 |
| Hours per week spent in practice |  |  |
| 1–4 | 0 | 1 |
| 5–9 | 7 | 11 |
| 10–14 | 9 | 7 |
| 15–19 | 0 | 0 |
| 20+ | 1 | 1 |
| Hours per week spent in games |  |  |
| 1–4 | 6 | 9 |
| 5–9 | 8 | 9 |
| 10–14 | 2 | 1 |
| 15–19 | 0 | 0 |
| 20+ | 1 | 1 |
| High school team achievements |  |  |
| Made playoffs % | 47.1 | 60 |
| Won championships % | 11.8 | 35 |

15 sports. This was notable, considering that the average age of survey respondents was 42 years old, which would indicate that many participants in our sample were at least 25 years post-high school. Our study provides preliminary evidence that individuals are reliable reporters of their high school sports-participation, and the survey presented here is one instrument that could be used for this purpose. However, this is a pilot study with small sample sizes and further research is warranted.

Of note, our results support the validity of self-reported retrospective assessment of the highest frequency and highest concussion risk sports evaluated in the present sample. The self-reported retrospective assessment of football, basketball, soccer, and wrestling participation all demonstrated excellent agreement with participation recorded in school yearbooks.

## Limitations

There are limitations of the present study that bear noting. The survey queries sports participation throughout childhood and adolescence. However, we were only able to evaluate the reliability of participants' reports of high school sports. Hence, the present data do not allow us to evaluate the reliability of retrospective reports of sports in earlier childhood. Although we collected information about the intensity of sports participation, we did not have corroborative yearbook data to evaluate the reliability of these responses. Some of the sports we evaluated have very small sample sizes. The sample was a convenience sample comprised of the authors' alumni networks, and not a representative sample of different types of schools across geographic regions. Hence it is possible that characteristics of the sample may have influenced our results in unanticipated ways.

Inconsistencies between survey responses and school records may have been due to memory errors on the part of survey respondents. There are two types of memory errors that would

result in inaccurate self-report: retrieval errors (or failure to report prior participation that did take place) and confabulations (or erroneous report of sports participation that did not occur). Inconsistencies due to memory failure would most likely show the pattern of higher sports participation rates indicated by school records as compared to survey responses, as confabulatory errors are relatively uncommon for the type of autobiographical information assessed in the present study—memory for sustained regular participation in a highly engaging experience, rather than a brief non-distinctive repetitive event (i.e. a single instance of taking a daily medication [16]). However, aside from cheerleading, cross country, skiing, and frisbee, the counts for self-reported participation in all other sports were greater than or equal to the corresponding participation counts obtained from school resources. This finding suggests that inconsistency between self-reported participation and school records may be due to errors in record keeping, rather than memory failures on the part of the survey respondents. These discrepancies could be due to inaccurate school records related to student absence on the day of team yearbook photos or errors in listing all athletes' names in photo captions.

## Future directions

Limitations notwithstanding, the present study supports the reliability of retrospective self-reported sports participation, particularly in the highest frequency, highest concussion risk sports. This survey may be an efficient way to incorporate youth sports participation into existing datasets for research purposes, providing an opportunity to better characterize the risks and benefits of youth participation in contact sports on cognitive, physical, and emotional health across the lifespan.

Further assessment of the reliability of this survey could be a next step to integrating it in a large, representative sample. In doing so, we can better understand the long-term implications of adolescent sports participation on trajectories of health as well as financial and psychosocial well-being. Future studies may seek to replicate the present findings in a larger sample, which would enable the opportunity to compare reliability across different sports. Our survey contains measures of sports-intensity (time spent in participation and competitive levels) but we were not able to evaluate their reliability. Sports-intensity is potentially important for investigators to consider, because it may be relevant to both psychological and physical effects of athletic participation that may bear on health risks and benefits [17,18]. Future work could evaluate the reliability of the sports-intensity using collateral interviews (e.g. coaches, parents), local newspaper reports, and school records.

There are a number of longitudinal data sets that have been following individuals from childhood/adolescence into later life. The addition of reliable instruments for retrospective self-report of youth sports participation to these data sets would provide an efficient and cost-effective method for examining the effects of childhood/adolescent sports participation on later life outcomes.

## Supporting information

**S1 Survey. Survey used for study.**
(PDF)

**S1 Dataset. Sports participation data.**
(XLSX)

**S2 Dataset. Cleaned participation data used for analysis.**
(XLSX)

**S1 Text. Dataset and survey question explanations.**
(RTF)

## Acknowledgments

We thank all the survey participants and special thanks to Susan Paterson, Tim Seminerio, Genevieve Strycharz, Stephen Loy, Timothy Hetrich and Corey Jones for their help with accessing school resources.

## Author Contributions

**Conceptualization:** Amanda R. Rabinowitz, Jordan Weiss, Sameer Deshpande, Reuben A. Buford May, Dylan S. Small.

**Data curation:** Steven Jin, Sameer Deshpande, Nitika Gupta.

**Formal analysis:** Amanda R. Rabinowitz, Jordan Weiss, Sameer Deshpande, Dylan S. Small.

**Funding acquisition:** Jordan Weiss, Nitika Gupta, Dylan S. Small.

**Methodology:** Steven Jin, Amanda R. Rabinowitz, Jordan Weiss, Sameer Deshpande, Nitika Gupta, Dylan S. Small.

**Resources:** Amanda R. Rabinowitz, Dylan S. Small.

**Supervision:** Dylan S. Small.

**Validation:** Sameer Deshpande.

**Writing – original draft:** Steven Jin.

**Writing – review & editing:** Steven Jin, Amanda R. Rabinowitz, Jordan Weiss, Sameer Deshpande, Reuben A. Buford May, Dylan S. Small.

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
