## [Decision Letter · Decision Letter 0]

13 Jan 2021

PONE-D-20-27226

Retrospective survey of youth sports participation: development and validation using school records

PLOS ONE

Dear Dr. Jin,

Thank you for submitting your manuscript to PLOS ONE. After careful consideration, we feel that it has merit but does not fully meet PLOS ONE’s publication criteria as it currently stands. Therefore, we invite you to submit a revised version of the manuscript that addresses the points raised during the review process.

Due to the current world situation is being difficult to find reviewers with time to help with the manuscripts. It has been a difficult way but we have a couple of reviewers that have made a great review. After their consideration, a major revision is needed. 

In addition, the journal policy implies to do the data fully available. Then, please, address this issue as well. 

We look forward to receiving your revised manuscript.

Kind regards,

Javier Brazo-Sayavera, Ph.D.

Academic Editor

PLOS ONE

Journal Requirements:

2. Please include additional information regarding the survey or questionnaire used in the study and ensure that you have provided sufficient details that others could replicate the analyses. For instance, if you developed a questionnaire as part of this study and it is not under a copyright more restrictive than CC-BY, please include a copy of the full version, in both the original language and English, as Supporting Information.

Reviewers' comments:

Reviewer's Responses to Questions

**Comments to the Author**

1. Is the manuscript technically sound, and do the data support the conclusions?

Reviewer #1: Partly

Reviewer #2: Yes

2. Has the statistical analysis been performed appropriately and rigorously? 

Reviewer #1: Yes

Reviewer #2: No

3. Have the authors made all data underlying the findings in their manuscript fully available?

Reviewer #1: No

Reviewer #2: No

4. Is the manuscript presented in an intelligible fashion and written in standard English?

Reviewer #1: Yes

Reviewer #2: Yes

5. Review Comments to the Author

Reviewer #1: This paper attempts to introduce and validate a retrospective survey of sports participation by utilising respondents self-reported sports participation data and validating responses using yearbook and online data bases. The introduction to the paper raised some interesting points, specifically around the investigation of physical and psychological risks of long-term sports participation, however no attempt seemed to be made to answer these questions There is also a lack of clarity relating to the methods section, particularly around the items used in the questionnaire – especially what was to be validated and what was not. The data set relating to intensity did not seem to be available. More broadly, it is felt that the aim of the paper and the associated research question is underdeveloped and unclear – the purpose and the need for the survey is questioned. As such, it is believed the paper is not currently suitable for publication. Below is a sectional review of the paper – it is hoped the authors find the more focused comments useful.

Introduction

The introduction begins with an opening line, and much of the opening paragraph, highlighting the detrimental effects of repeated head injury, however the authors do not follow this sporting and social dilemma within the survey investigation. The need to undertake work to survey longitudinal data relating to head trauma and sports participation versus the dilemma of the positive effects of sports participation were established on page 1 and 2 – however this did not seem to continue through the remaining sections of the paper.

For example, the authors need to clarify why participants were not asked if they suffered concussion during their time at high school. Why was this question not included in the survey – especially when they included questions relating to intensity, which in the end could not be validated?

Page 3 paragraph 2, line 1-5: A decline in sports participation was noted, however these figures were not substantiated. For example, if a drop in 2018-19 was recorded what were the previous years figures?

Page 4, 1st paragraph, line 5-7: mixed findings from previous research (and there seems to be numerous studies available) warrants further research investigating the impact of high school sports on concurrent and later-life outcomes. This process of investigation does not seem to be associated with the aims and survey development outlined? It was felt the introduction takes the reader on an important journey and raises important issues, but the path does not lead anywhere.

It may be useful for the authors to utilise the introduction in a more focused way – i.e. what the gap in the current literature is and where this paper fits within that. Within this, it would also be useful to offer more detail on how existing survey tools and questionnaires require this improvement.

Methods

Survey development:

Line 1: What survey items were developed by the authors and what informed their inclusion? How did this compare with other surveys? What are the differences – this will assist in understanding why we need this survey?

Line 1 and 2: Sports were included for the potential of concussion/risk to hit on head; however, this was not a question in the survey (see above point).

How many items were included in the survey?

Results:

In the discussion section it is noted this is indeed a pilot study. It would have been useful to highlight this in the introduction or methods. This can account for the low numbers associated with the paper.

Also, as the number of items were not included in the survey development section it is difficult to comment on the sample size. However, it seems, for some sports, a high agreement scores were achieved for low participant numbers.

Further clarity is required regarding what participants were agreeing to (the items) in order to achieve full recall after at least 20+ years out of high school. For example, in the ‘weakness’ section of the discussion it states that the intensity data could not be validated – therefore was only a single binary question relating to type of sport and confirmation of participation requested? This is unclear.

Did the information contained in High School year books assist in item formation? If it did not, the study assumes all high school yearbooks will contain comparative data. This may be the case if the items used for validation was a simple yes/no to the sports the respondents participated in – but this is not clear.

Discussion:

Page 10 line 1: This feels like quite an optimistic statement, mostly due to sample sizes when breaking the findings down to specific sports.

In the limitations part (there is no specific limitations section) it is noted that the study provides no way to evaluate the validity of the sports intensity measures – this is a crucial omission, as it does not provide the opportunity to delve deeper into the psychological and physical effects of athletic participation. In addition, the first paragraphs of the study seem to outline what data could or would have been useful to collect, rather than focusing on what was collected. In order to rectify this the authors may wish to consider a restructure the paper.

Also, in the limitations (weakness) section the use of a convenience sample was described – this should be noted and acknowledged in the methods section.

In the final paragraph there is first mention of ‘pilot study’ – this should have been detailed in the methods section.

It may be useful for the authors to restructure the discussion section by having more specific sections. For example, the authors offer some discussion on possible applications, however this could be more focused. In addition, a separate limitations section would also be useful.

Reviewer #2: Big picture

The topic the authors address is a relevant issue in the field and the authors should be commended on their attempt to validate a measure of previous sport participation. The study over all was well written and presented, however there are some concerns that need to be addressed.

Specifically the use of Cronbach’s alpha and Cohen’s kappa as appropriate for this study i needed. Previously internal consistency is shown as correlations between different items or sub scales on the same measure not between measures, and inter-rater reliability is between separate observers, RA's etc. using the same measure, again not between measures. Justification for using these methods for a between measures analysis is needed.

"For example, in addition to summary statistics, the data points behind means, medians and variance measures should be available." Any data on hours, years, measures of intensity, as the authors labeled it, is missing from the submission.

Details

Introduction

I found it difficult to see the connection of the introduction (particularly the first three paragraphs) to the purpose and results of the paper. While the introduction was clear and well written, the link between the focus on contact sports and concussions as the supporting literature and the validation of the tool itself was missing. I would suggest cutting the first section on contact sports and CTE unless authors can explicitly link it to the validating of the measure on all sport participation. There is literature that highlights a need for a valid measure of previous sport participation, and this would be better suited for the introduction. Further the authors highlight the optimal ways of collecting data (prospective, longitudinal) but then settle on a retrospective method, while they provide some information as to why the other methods aren’t often used, there is no explanation why their retrospective method is used or why it is beneficial.

Sample Methods

The authors need to explain why photo and yearbook rosters were used over other methods. The authors themselves point out that people could often be away for picture day and year books could have incorrect or missing names. Additionally, depending when the photos were taken athletes could have quit the team soon after and therefore the information would not match. It is an interesting and novel approach, but further justification is needed.

Statistical analysis

“self-report used as one source of data and the school record used as another”

The authors should explain why these atypical approaches are suitable for their purposes, especially on top of the approach of percent agreement. Are these approaches appropriate to use for inter-rater reliability or internal consistency? If so, the authors need to provide examples of when this has been done in the past or why this should be considered an appropriate use of Cronbach’s alpha and Cohen’s kappa.

Results

The authors should provide justification for removing participants who took a long time to complete. Was completing the survey in one time session required? Why was interruption a problem ?

The purpose of the paper was to validate a measure of sport participation, yet there is not indication of an effort to validate measure of intensity. The authors collected information on hours, year and achievements, but this section was not validated? I suggest the authors explain why they did not validate this section, show that is was validated or remove it from the paper. The authors did mention that the year-book does not provide measures on intensity but there are other methods for validating this information, or if it does not fit with the year-book method then it should be removed.

Table 1.

Title is misleading, “Interrater Reliability for Survey Responses and School Resources” as this table also shows, internal consistency and precent agreement?

Discussion

Again the authors highlight concerns around using the yearbook information to validate measures “inconsistency between self-reported participation and school records may be due to errors in record keeping, rather than memory failures on the part of the survey respondents”

and therefore the authors need to make a stronger argument for use of yearbooks.

The discussion highlights the need for information on intensity, but the study fails to validate this information. Such a focus on intensity and consequences of intensity in the discussion seems out of place when measures of intensity were not validated.

Supplemental information

The authors provide the data for the coding and analysis of sports but there is not data with intensity information ?

Overall, the study provides a novel approach to validating a measure of high school sport participation and is an important endeavour, however the paper requires further explanations and some reorganizing to be suitable for publication.

6. PLOS authors have the option to publish the peer review history of their article (what does this mean?). If published, this will include your full peer review and any attached files.

Reviewer #1: No

Reviewer #2: **Yes: **Alexandra Mosher

---

## [Author Response · Author response to Decision Letter 0]

30 Mar 2021

Reviewer 1 Comment 1: This paper attempts to introduce and validate a retrospective survey of sports participation by utilising respondents self-reported sports participation data and validating responses using yearbook and online databases. The introduction to the paper raised some interesting points, specifically around the investigation of physical and psychological risks of long-term sports participation, however no attempt seemed to be made to answer these questions There is also a lack of clarity relating to the methods section, particularly around the items used in the questionnaire – especially what was to be validated and what was not. The data set relating to intensity did not seem to be available. More broadly, it is felt that the aim of the paper and the associated research question is underdeveloped and unclear – the purpose and the need for the survey is questioned. As such, it is believed the paper is not currently suitable for publication. Below is a sectional review of the paper – it is hoped the authors find the more focused comments useful.

Thanks for the useful comments. We have edited the paper to clarify that the aim of the paper is to introduce a survey of retrospective sports participation and to validate aspects of the survey. We have edited the abstract and introduction to clarify these aims. For example, we have rewritten the first two lines of the abstract to be, “Many youth participate in sports and it is of interest to understand the impact of youth sports participation on later-life outcomes. However, prospective studies take a long time to complete and retrospective studies may be more practical and time-efficient to address some questions.” We have edited the introduction and discussion to clarify that the survey is intended to help with investigating physical and psychological risks and benefits of sports participation, but we do not answer questions about the risks and benefits in this paper. We have rewritten the methods section to clarify which responses in the survey were validated and which were not. We have made the data set related to intensity available. 

Reviewer 1 Comment 2: The introduction begins with an opening line, and much of the opening paragraph, highlighting the detrimental effects of repeated head injury, however the authors do not follow this sporting and social dilemma within the survey investigation. The need to undertake work to survey longitudinal data relating to head trauma and sports participation versus the dilemma of the positive effects of sports participation were established on page 1 and 2 – however this did not seem to continue through the remaining sections of the paper. For example, the authors need to clarify why participants were not asked if they suffered concussion during their time at high school. Why was this question not included in the survey – especially when they included questions relating to intensity, which in the end could not be validated?

We agree with the reviewer that the introduction could be better structured to set up the aims of the study. In response to this comment, we have restructured the introduction to lead with this discussion of youth sports participation, which is the subject of the survey. We did not include survey items on history of brain injury/concussion because validated measures for assessing history of brain injury history are already available. We now include this information in the introduction. 

Reviewer 1 Comment 3: Page 3 paragraph 2, line 1-5: A decline in sports participation was noted, however these figures were not substantiated. For example, if a drop in 2018-19 was recorded what were the previous years figures?

The number of participants in high school sports declined from 7,980,886 in 2017-2018 to 7,937,491 in 2018-2019. These figures come from the National Federation of State High School Associations. Our revised introduction now includes these figures.

Reviewer 1 Comment 4. Page 4, 1st paragraph, line 5-7: mixed findings from previous research (and there seems to be numerous studies available) warrants further research investigating the impact of high school sports on concurrent and later-life outcomes. This process of investigation does not seem to be associated with the aims and survey development outlined? It was felt the introduction takes the reader on an important journey and raises important issues, but the path does not lead anywhere.

It may be useful for the authors to utilise the introduction in a more focused way – i.e. what the gap in the current literature is and where this paper fits within that. Within this, it would also be useful to offer more detail on how existing survey tools and questionnaires require this improvement.

We have now restructured the introduction per the reviewers’ suggestion, putting discussion of youth sports participation up front, and also adding detail on the current measures available for assessing prior brain injury. In the penultimate paragraph of the introduction, we directly state the gap in the current literature, and how the survey presented in the paper fills that gap. 

Reviewer 1 Comment 5: What survey items were developed by the authors and what informed their inclusion? How did this compare with other surveys? What are the differences – this will assist in understanding why we need this survey? Sports were included for the potential of concussion/risk to hit on head; however, this was not a question in the survey (see above point). How many items were included in the survey?

We developed our survey using a series of blocks to assess sports participation at various stages of one's education. This included a block for grades K-6, grades 7-8, grades 9-12, and college. We structured the survey this way to assess the timing and duration of sports participation while inherently collecting information about competitiveness (eg, participation in college sports is likely to be more competitive than grades 7-8 or high school [grades 9-12]). The four blocks (K-6, grades 7-8, grades 9-12, and college) were identical in the sense that the same series of questions and response options were provided. In addition, we included an introductory block to collect demographic information on the respondent (e.g., age, gender, race, ethnicity, educational attainment, and where they attended high school).

The sports included in our survey were derived, in part, from the high school athletics participation survey conducted by the National Federation of State High School Associations. We also took cues from the Life History Survey 2017 of the Health and Retirement Study which assessed whether respondents participated in sports during junior/middle and high school. However, one potential limitation of the aforementioned survey is that it has not been validated. Thus, we developed the survey items for our survey by drawing from these existing surveys in addition to collecting detailed information about high school attendance history which allowed us to validate survey responses using historical yearbook data.

In total, there were 545 survey items but these were based on branching logic; if an individual participated in every listed support at each time point, they would have responded to 545 queries.

Reviewer 1 Comment 6: In the discussion section it is noted this is indeed a pilot study. It would have been useful to highlight this in the introduction or methods. This can account for the low numbers associated with the paper. Also, as the number of items were not included in the survey development section it is difficult to comment on the sample size. However, it seems, for some sports, a high agreement scores were achieved for low participant numbers. Further clarity is required regarding what participants were agreeing to (the items) in order to achieve full recall after at least 20+ years out of high school. For example, in the ‘weakness’ section of the discussion it states that the intensity data could not be validated – therefore was only a single binary question relating to type of sport and confirmation of participation requested? This is unclear. Did the information contained in High School year books assist in item formation? If it did not, the study assumes all high school yearbooks will contain comparative data. This may be the case if the items used for validation was a simple yes/no to the sports the respondents participated in – but this is not clear.

We have now highlighted the fact that our study is indeed a pilot study earlier in the abstract and introduction of our paper. We have also added an acknowledgement of the small sample sizes -- see last sentence of first paragraph of the Discussion. The high school yearbooks were not used to design our survey items. Rather, we used the yearbook to validate the survey results. 

We have provided further clarity about the survey and what respondents were agreeing to by adding a subsection “Survey” to the Methods Section of our revised manuscript in which we write “In our survey, subjects were asked to select the sports in which they participated from a list of radio buttons. The survey used a series of blocks to assess sports participation at various stages of one's education. This included a block for grades K-6, grades 7-8, grades 9-12, and college. For each block corresponding to a specific period in the life course, respondents were asked to select the sports in which they participated from a list of radio buttons. Using this, binary variables were constructed for participation in each sport during high school. Respondents also indicated whether, at each specific period, they had any physical limitations that prevented them from participating in sports. For each selected sport at each time period, respondents were asked to report (using radio buttons) the specific grades in which they participated, whether their team (a) traveled for games or tournaments, (b) participated in playoffs, (c) won a championship, or (d) none of the above. Respondents also indicated how many hours per week (0 hours, 1-4 hours, 5-9 hours, 10-14 hours, 15-19 hours, 20+ hours) they spent practicing and playing games for each specific sport, in addition to the position(s) they played.” 

Reviewer 1 Comment 7: Page 10 line 1: This feels like quite an optimistic statement, mostly due to sample sizes when breaking the findings down to specific sports. In the limitations part (there is no specific limitations section) it is noted that the study provides no way to evaluate the validity of the sports intensity measures – this is a crucial omission, as it does not provide the opportunity to delve deeper into the psychological and physical effects of athletic participation. In addition, the first paragraphs of the study seem to outline what data could or would have been useful to collect, rather than focusing on what was collected. In order to rectify this the authors may wish to consider a restructure the paper. Also, in the limitations (weakness) section the use of a convenience sample was described – this should be noted and acknowledged in the methods section. In the final paragraph there is first mention of ‘pilot study’ – this should have been detailed in the methods section. It may be useful for the authors to restructure the discussion section by having more specific sections. For example, the authors offer some discussion on possible applications, however this could be more focused. In addition, a separate limitations section would also be useful.

Thanks for these helpful comments. We have qualified the statement originally on Page 10, line 1: “Our study provides preliminary evidence that individuals are reliable reporters of their high school sports-participation, and the survey presented here is one instrument that could be used for this purpose. However, this is a pilot study with small sample sizes and further research is warranted.”

We have edited the first sentence of the “Sample Collection” section of the Methods section to acknowledge that this is a pilot study with a convenience sample: “A pilot study was conducted to assess the reliability of the survey by constructing a convenience sample that distributed the survey through alumni e-mail lists and social media groups from high schools where authors (D.S. and A.R) are alumni.”

We have restructured the discussion to have more specific sections and provided a more focused direction of possible applications in our “Future Directions” section. 

Reviewer 2 Comment 1: The topic the authors address is a relevant issue in the field and the authors should be commended on their attempt to validate a measure of previous sport participation. The study overall was well written and presented, however there are some concerns that need to be addressed. Specifically the use of Cronbach’s alpha and Cohen’s kappa as appropriate for this study is needed. Previously internal consistency is shown as correlations between different items or sub scales on the same measure not between measures, and inter-rater reliability is between separate observers, RA's etc. using the same measure, again not between measures. Justification for using these methods for a between measures analysis is needed. Any data on hours, years, measures of intensity, as the authors labeled it, is missing from the submission.

We appreciate the reviewer’s overall assessment of our work. We concede that our use of Cronbach’s alpha and Cohen’s kappa is somewhat nonstandard. In our revised manuscript, we have removed the analysis based on Cohen’s kappa. We have, however, retained the analysis based on Cronbach’s alpha for the following reason. Essentially, we have two ostensible measurements of the single construct, sports participation. Computing Cronbach’s alpha allows us to assess to the extent to which these two measures really are measuring the same thing. We have included this justification in our revised manuscript. We will make a de-identified data file available. 

Reviewer 2 Comment 2: I found it difficult to see the connection of the introduction (particularly the first three paragraphs) to the purpose and results of the paper. While the introduction was clear and well written, the link between the focus on contact sports and concussions as the supporting literature and the validation of the tool itself was missing. I would suggest cutting the first section on contact sports and CTE unless authors can explicitly link it to the validating of the measure on all sport participation. There is literature that highlights a need for a valid measure of previous sport participation, and this would be better suited for the introduction. Further the authors highlight the optimal ways of collecting data (prospective, longitudinal) but then settle on a retrospective method, while they provide some information as to why the other methods aren’t often used, there is no explanation why their retrospective method is used or why it is beneficial.

We have now restructured the introduction (see R1.2), cutting some of the background on CTE, and adding a more focused discussion of how the present study fills gaps in the literature. We have also added the following to the final paragraph of the introduction, 

“Retrospective assessment of youth sports participation has the advantage of allowing investigators to capitalize on existing longitudinal datasets that have already tracked cognitive and health outcomes into later-life, when some of the deleterious effects of brain injury are thought to manifest.”

Reviewer 2 Comment 3. The authors need to explain why photo and yearbook rosters were used over other methods. The authors themselves point out that people could often be away for picture day and year books could have incorrect or missing names. Additionally, depending when the photos were taken athletes could have quit the team soon after and therefore the information would not match. It is an interesting and novel approach, but further justification is needed.

We considered several other potential sources against which to validate the self-reported sports participation data. Unfortunately many of them, such as retrospective reports from parents, siblings, coaches, and friends, are subject to the same potential recall biases as the self-reports themselves. In the absence of prospectively collected participation data, yearbooks and school records offer a nearly-contemporaneous source of sports participation that is not subject to recall bias.

That being said, the referee is correct that yearbooks are not 100% reliable. Nevertheless, our preliminary results suggest that there may be considerable agreement between retrospective self-reports and yearbook records. 

Reviewer 2 Comment 4: The authors should explain why these atypical approaches are suitable for their purposes, especially on top of the approach of percent agreement. Are these approaches appropriate to use for inter-rater reliability or internal consistency? If so, the authors need to provide examples of when this has been done in the past or why this should be considered an appropriate use of Cronbach’s alpha and Cohen’s kappa.

In a certain sense, we have two items (yearbook record and self-report) that purport to measure the same thing (sports participation). As a result, we argue that it is useful to consider the internal consistency of these two items and to report the value of Cronbach’s alpha. 

Reviewer 2 Comment 5. The authors should provide justification for removing participants who took a long time to complete. Was completing the survey in one time session required? Why was interruption a problem?

We thank the reviewer for their comments. We wish to make a point of clarification regarding the removal of participants. These respondents were not removed from the analytic sample; we only excluded them to calculate the adjusted mean survey completion time after accounting for outliers. However, we agree that this may be a point of confusion in the manuscript. As such, we removed this comment from the main text.

Reviewer 2 Comment 6: The purpose of the paper was to validate a measure of sport participation, yet there is no indication of an effort to validate measure of intensity. The authors collected information on hours, year and achievements, but this section was not validated? I suggest the authors explain why they did not validate this section, show that it was validated, or remove it from the paper. The authors did mention that the year-book does not provide measures on intensity but there are other methods for validating this information, or if it does not fit with the year-book method then it should be removed.

We considered other methods for validating intensity items. Detailed school reports of participation intensity were not available for this purpose. We considered collateral report from either parents or other school sources, however, these reports would have been burdensome to acquire, and we were not at all confident that they would be appropriate measures for validating this information-- that is, the retrospective reports from collaterals would also be subject to memory biases. Hence, we determined that there was no appropriate comparison standard for evaluating the validity of these items in the present study.

Despite this limitation, we have decided to include the intensity data in the manuscript. These data highlight an aspect of the survey that captures more granular information on sports participation, which we believe is a potential strength of the survey. The predictive validity of these data could be established in future work examining relationships between sports participation intensity and outcomes of interest. Such an examination is outside of the scope of the current work, and an important direction for future research. 

Reviewer 2 Comment 7: Table 1: Title is misleading, “Interrater Reliability for Survey Responses and School Resources” as this table also shows, internal consistency and percent agreement?

Thank you for pointing this out. We will rename the table “Percent Agreement and Internal Consistency for Survey Responses and School Resources”

Reviewer 2 Comment 8: Again the authors highlight concerns around using the yearbook information to validate measures “inconsistency between self-reported participation and school records may be due to errors in record keeping, rather than memory failures on the part of the survey respondents” and therefore the authors need to make a stronger argument for use of yearbooks. The discussion highlights the need for information on intensity, but the study fails to validate this information. Such a focus on intensity and consequences of intensity in the discussion seems out of place when measures of intensity were not validated.

We appreciate the reviewer’s pointing this out. As we argue in response to Reviewer 2 Comment 3, yearbooks are not subject to the same biases as self-report or collateral interviews with parents, friends, and school officials. The relatively high agreement observed in our pilot study suggests that there may not be substantial errors in record keeping. We have reduced the emphasis on intensity in the discussion of the revised manuscript.

Reviewer 2 Comment 9: The authors provide the data for the coding and analysis of sports but there is not data with intensity information ?

We have now included a data file that contains intensity information such as years played, hours spent per week, and team/individual achievements.

---

## [Decision Letter · Decision Letter 1]

27 May 2021

PONE-D-20-27226R1

Retrospective survey of youth sports participation: development and validation using school records

PLOS ONE

Dear Dr. Jin,

Thank you for submitting your manuscript to PLOS ONE. After careful consideration, we feel that it has merit but does not fully meet PLOS ONE’s publication criteria as it currently stands. Therefore, we invite you to submit a revised version of the manuscript that addresses the points raised during the review process.

Pay attention to reviewer's comments in order to address all the specific issues. We are sorry for the delay but it is too difficult to find reviewers with enough expertise for this review. 

We look forward to receiving your revised manuscript.

Kind regards,

Javier Brazo-Sayavera, Ph.D.

Academic Editor

PLOS ONE

Journal Requirements:

Reviewers' comments:

Reviewer's Responses to Questions

**Comments to the Author**

1. If the authors have adequately addressed your comments raised in a previous round of review and you feel that this manuscript is now acceptable for publication, you may indicate that here to bypass the “Comments to the Author” section, enter your conflict of interest statement in the “Confidential to Editor” section, and submit your "Accept" recommendation.

Reviewer #2: (No Response)

Reviewer #3: All comments have been addressed

2. Is the manuscript technically sound, and do the data support the conclusions?

Reviewer #2: Partly

Reviewer #3: Partly

3. Has the statistical analysis been performed appropriately and rigorously? 

Reviewer #2: No

Reviewer #3: Yes

4. Have the authors made all data underlying the findings in their manuscript fully available?

Reviewer #2: Yes

Reviewer #3: Yes

5. Is the manuscript presented in an intelligible fashion and written in standard English?

Reviewer #2: Yes

Reviewer #3: Yes

6. Review Comments to the Author

Reviewer #2: I appreciated that the authors took the time to respond to all comments and to revise some aspects of the paper, however there are still some concerns that need to be addressed.

BIG PICTURE

The main concern is still around the use of Cronbachs alpha. If you are only looking at number of sports from two different sources, then technically you only have two items. There are concerns around using cronbach’s alpha with only two items See, Eisinga, R., Te Grotenhuis, M., & Pelzer, B. (2013). The reliability of a two-item scale: Pearson, Cronbach, or Spearman-Brown?. International journal of public health, 58(4), 637-642. The authors might consider only using percent agreement in this study.

There is a concern around the use of the word validity and validation in the study. Finding reliability does not necessarily mean that a measure is valid. It can be misleading to use the word validity when there was no measure of validity in the study. This may be incorrect, so I leave it up to the editors discretion.

Survey development / Survey

I am unsure how the sports included in the study were chosen at the authors state two different reasons. “The sports included in the survey were selected for their popularity at the high school/collegiate level and the potential risk for a hit on the head or concussion” and

“The specific 15 sports (listed in Table 1) included in our survey were selected based on sports participation surveys conducted by the National Federation of State High School Associations.”

Please clarify which was the correct criteria for inclusion.

Statistical analysis

Please edit this sentence it says context twice and reads awkwardly - “ In our context, high values of Cronbach’s alpha in our context indicate that the retrospective self-reports and yearbook records are roughly measuring the same construct, sports participation”

Discussion

The last sentence mentions of ‘all three metrics’ which three are you referring to? Or do you mean both metrics?

Limitations

I do not know what you mean by “validity sample” in the sentence “ The validity sample was a convenience sample comprised of the authors’ alumni networks”

Future directions

Again the use of validity seems misleading ‘Limitations not withstanding, the present study supports the validity of retrospective self-reported sports participation, particularly in the highest frequency, highest concussion risk sports.”

I believe you supported the reliability of the measure?

Reviewer #3: Thank you for responding to reviewers' recommendations thoroughly. I do think that in the absence of actual longitudinal studies, this retrospective method with some improvements could fill the current gap. I would remove all associations with contact sport reference, and keep it broad and just concentrate on validation of the survey as a method, confirming that long term sport participation need to be investigated for its benefits too, precisely because of the large drop in participation observed in recent years. Just a small issue of typos - participants' not participant's. Check manuscript throughout.

7. PLOS authors have the option to publish the peer review history of their article (what does this mean?). If published, this will include your full peer review and any attached files.

Reviewer #2: **Yes: **Alexandra Mosher

Reviewer #3: **Yes: **Dr Erika Borkoles

---

## [Author Response · Author response to Decision Letter 1]

17 Jul 2021

Reviewer 2 Comment 1: BIG PICTURE

The main concern is still around the use of Cronbachs alpha. If you are only looking at number of sports from two different sources, then technically you only have two items. There are concerns around using cronbach’s alpha with only two items See, Eisinga, R., Te Grotenhuis, M., & Pelzer, B. (2013). The reliability of a two-item scale: Pearson, Cronbach, or Spearman-Brown?. International journal of public health, 58(4), 637-642. The authors might consider only using percent agreement in this study.

Response: We are grateful for the helpful references. We agree with the referee that Cronbach’s alpha may be inappropriate in this setting and we have removed this analysis from our revised manuscript. We now report only percent agreement between the yearbook record and survey self-report.

Reviewer 2 Comment 2: There is a concern around the use of the word validity and validation in the study. Finding reliability does not necessarily mean that a measure is valid. It can be misleading to use the word validity when there was no measure of validity in the study. This may be incorrect, so I leave it up to the editors discretion.

Response: Thank you for bringing up the distinction between reliability and validity. We have replaced the word “validity” with “reliability” throughout the manuscript and retitled the manuscript “The sample was a convenience sample comprised of the authors’ alumni networks, and not a representative sample of different types of schools across geographic regions.”

Review 2 Comment 3: Survey development / Survey

I am unsure how the sports included in the study were chosen at the authors state two different reasons. “The sports included in the survey were selected for their popularity at the high school/collegiate level and the potential risk for a hit on the head or concussion” and

“The specific 15 sports (listed in Table 1) included in our survey were selected based on sports participation surveys conducted by the National Federation of State High School Associations.”

Please clarify which was the correct criteria for inclusion.

Response: Thanks for pointing out this discrepancy. We have clarified the correct criteria for inclusion as “The specific 15 sports (listed in Table 1) included in our survey were selected based on sports participation surveys conducted by the National Federation of State High School Associations as well as knowledge of other sports that were popular at the schools where the pilot study described in the next section was conducted.”

Reviewer 2 Comment 4: Statistical analysis

Please edit this sentence it says context twice and reads awkwardly - “ In our context, high values of Cronbach’s alpha in our context indicate that the retrospective self-reports and yearbook records are roughly measuring the same construct, sports participation”

Response: Thanks for mentioning this. We have removed this sentence because we have removed the use of Cronbach’s alpha. 

Reviewer 2 Comment 5: Discussion

The last sentence mentions of ‘all three metrics’ which three are you referring to? Or do you mean both metrics?

Response: Thanks for pointing this out. We edited this sentence to “Of note, our results support the validity of self-reported retrospective assessment of the highest frequency and highest concussion risk sports evaluated in the present sample. The self-reported retrospective assessment of football, basketball, soccer and wrestling participation all demonstrated excellent agreement with participation recorded in school yearbooks”

Reviewer 2 Comment 6: Limitations

I do not know what you mean by “validity sample” in the sentence “ The validity sample was a convenience sample comprised of the authors’ alumni networks”

Response: Thanks for bringing this up. We removed the word “validity” so that the sentence is now “The validity sample was a convenience sample comprised of the authors’ alumni networks, and not a representative sample of different types of schools across geographic regions.”

Reviewer 2 Comment 7: Future directions

Again the use of validity seems misleading ‘Limitations not withstanding, the present study supports the validity of retrospective self-reported sports participation, particularly in the highest frequency, highest concussion risk sports.”

I believe you supported the reliability of the measure?

Response: Thanks for bring this up. We have replaced “validity” with “reliability” in the sentence so it now reads: “Limitations not withstanding, the present study supports the reliability of retrospective self-reported sports participation, particularly in the highest frequency, highest concussion risk sports.”

Reviewer 3 Comment 1: Thank you for responding to reviewers' recommendations thoroughly. I do think that in the absence of actual longitudinal studies, this retrospective method with some improvements could fill the current gap. I would remove all associations with contact sport reference, and keep it broad and just concentrate on validation of the survey as a method, confirming that long term sport participation need to be investigated for its benefits too, precisely because of the large drop in participation observed in recent years. 

Response: We have removed the paragraph reviewing the literature on contact sports and risk of traumatic brain injury. Now the introduction is more broadly focused on the long-term implications of sports participation. 

Reviewer 3 Comment 2: Just a small issue of typos - participants' not participant's. Check manuscript throughout.

Response: Thanks for pointing this out. We have replaced participant’s with participants’ throughout the manuscript.

---

## [Decision Letter · Decision Letter 2]

3 Sep 2021

Retrospective survey of youth sports participation: development and validation using school records

PONE-D-20-27226R2

Dear Dr. Jin,

We’re pleased to inform you that your manuscript has been judged scientifically suitable for publication and will be formally accepted for publication once it meets all outstanding technical requirements.

Kind regards,

Javier Brazo-Sayavera, Ph.D.

Academic Editor

PLOS ONE

Additional Editor Comments (optional):

Please, review the whole text to correct the typos.

Reviewers' comments:

Reviewer's Responses to Questions

**Comments to the Author**

1. If the authors have adequately addressed your comments raised in a previous round of review and you feel that this manuscript is now acceptable for publication, you may indicate that here to bypass the “Comments to the Author” section, enter your conflict of interest statement in the “Confidential to Editor” section, and submit your "Accept" recommendation.

Reviewer #2: All comments have been addressed

Reviewer #3: All comments have been addressed

2. Is the manuscript technically sound, and do the data support the conclusions?

Reviewer #2: Yes

Reviewer #3: Yes

3. Has the statistical analysis been performed appropriately and rigorously? 

Reviewer #2: Yes

Reviewer #3: Yes

4. Have the authors made all data underlying the findings in their manuscript fully available?

Reviewer #2: Yes

Reviewer #3: Yes

5. Is the manuscript presented in an intelligible fashion and written in standard English?

Reviewer #2: Yes

Reviewer #3: No

6. Review Comments to the Author

Reviewer #2: (No Response)

Reviewer #3: The authors satisfactorily addressed the reviewers' comments. There are a lot of typos in the text. Please do address these.

7. PLOS authors have the option to publish the peer review history of their article (what does this mean?). If published, this will include your full peer review and any attached files.

Reviewer #2: No

Reviewer #3: **Yes: **Erika Borkoles

---

## [Editor Report · Acceptance letter]

9 Sep 2021

PONE-D-20-27226R2 

Retrospective survey of youth sports participation: development and  assessment of reliability using school records 

Dear Dr. Jin:

I'm pleased to inform you that your manuscript has been deemed suitable for publication in PLOS ONE. Congratulations! Your manuscript is now with our production department. 

Kind regards, 

on behalf of

Dr. Javier Brazo-Sayavera 

Academic Editor

PLOS ONE